# A Few-Shot Approach for Relation Extraction Domain Adaptation using Large Language Models

Vanni Zavarella[1,*], Juan Carlos Gamero-Salinas[2] and Sergio Consoli[3]

[1]*Department of Mathematics and Computer Science, University of Cagliari, Cagliari, Italy*
[2]*Institute of Data Science and Artificial Intelligence (DATAI), Universidad de Navarra, Pamplona, Spain*
[3]*European Commission, Joint Research Centre (DG JRC), Ispra (VA), Italy*

### Abstract

Knowledge graphs (KGs) have been successfully applied to the analysis of complex scientific and technological domains, with automatic KG generation methods typically building upon relation extraction models capturing fine-grained relations between domain entities in text. While these relations are fully applicable across scientific areas, existing models are trained on few domain-specific datasets such as SciERC and do not perform well on new target domains. In this paper, we experiment with leveraging in-context learning capabilities of Large Language Models to perform schema-constrained data annotation, collecting in-domain training instances for a Transformer-based relation extraction model deployed on titles and abstracts of research papers in the Architecture, Construction, Engineering and Operations (AECO) domain. By assessing the performance gain with respect to a baseline Deep Learning architecture trained on off-domain data, we show that by using a few-shot learning strategy with structured prompts and only minimal expert annotation the presented approach can potentially support domain adaptation of a science KG generation model.

### Keywords

Large Language Models, Knowledge Graphs, Few-shot Learning, Relation Extraction, Data Augmentation

## 1. Introduction

Knowledge graphs (KGs) [1] have proved effective for representing research knowledge discussed in scientific papers and patents across several different domains [2, 3, 4]. New generation "scientific KGs" have moved from representing purely bibliographic information of research publications to support the construction of extensive networks of machine-readable information about entities and relationships pertaining to a certain domain, enabling fine-grained semantic queries over large scientific text collections such as: "retrieve all methods that are used for Indoor Air Remediation in the time range *T*".

Therefore, they can support downstream analytical services like technology trend analysis. For example, [5] uses the statistics of relation triples of type *<Method;Used-for;Task>* automatically extracted from paper abstracts to reconstruct historical trends of the top applications of target methods such as "neural networks" in different areas like speech recognition and computer vision.

*Workshop at KDD 2024 on Deep Learning and Large Language Models for Knowledge Graphs, 25-29 August, Barcelona.*
*Corresponding author.

✉ vanni.zavarella@unica.it (V. Zavarella); jgamero@unav.es (J. C. Gamero-Salinas); sergio.consoli@ec.europa.eu (S. Consoli)

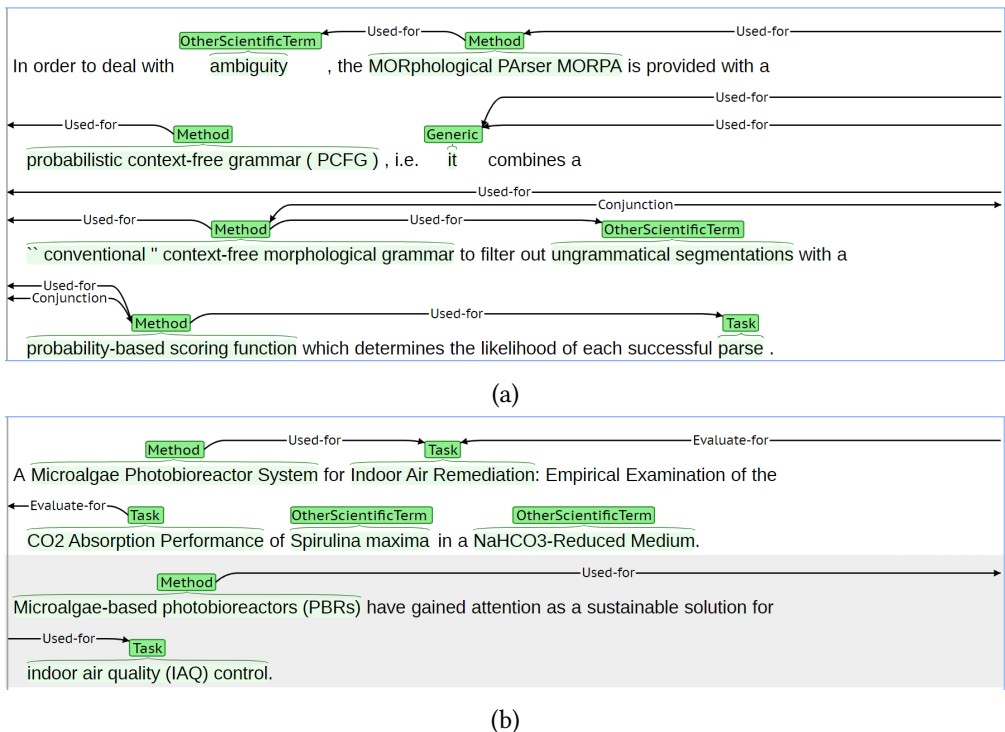

**Figure 1:** Sample Entity and Relation annotation from the SciERC dataset (a) and sample annotation from an AECO paper abstract ([7]), following the SciERC annotation schema.

Methods for automatic generation of scientific KGs typically build upon training supervised Relation Extraction (RE) models for capturing fine-grained relations between scientific entities in text [6]. In the scholarly domain, the entity and relation specifications for this task are defined by the SciERC initiative [5][1] and are fully applicable across scientific areas. As part of an ongoing endeavour on innovation intelligence analytics for the Architecture, Engineering, Construction and Operation (AECO) industry, we applied SciERC guidelines to annotate domain-specific instances of scientific entities (e.g. *Task*, *Method*, *Metrics*) and their relations (e.g. *Used-for*, *Evaluate-for*). In Figure 1 we show a few sentence-level annotations of SciERC entity and relation types respectively from an NLP paper abstract from the SCIERC dataset (a) and from an AECO research paper annotated via the same tagging schema (b).

One can see that the semantic schema is perfectly portable to the AECO domain. However, state-of-the-art models for scholarly Relation Extraction have been trained on the SciERC dataset comprising AI/ML articles and do not perform well on new target domains such as AECO[8, 9]. Moreover, manually annotating training data for this domain is a costly and time-consuming strategy that does not scale well.

Therefore, we experiment on an empirical solution leveraging in-context learning capabilities of Large Language Models (LLMs) [10, 11] to perform schema-constrained data annotation, generating in-domain training instances for a baseline Relation Extraction architecture from only

---

[1]https://nlp.cs.washington.edu/sciIE/annotation_guideline.pdf

a few manually designed sentence annotation examples, together with explicit task instructions. Then, we test different training configurations of the model on a small test set of titles and abstracts of AECO research papers and compare the performance with a baseline trained on out-of-domain data.

Research on techniques for distilling knowledge from pre-trained LLMs to downstream NLP tasks is currently highly active [12]. Prompt tuning approaches, that translate the target downstream tasks to a masked language modeling problem have been applied to Relation Classification [13, 14, 15]. Similarly to [15], we combine few-shot examples prompt with rich schema information to elicit LLMs' comprehension of the RE task. However, we explicitly formulate the task as data annotation to the LLM.

Finally, the presented approach is in line with the view from [12] that leveraging few-shot learning capabilities of LLMs for optimizing local, lower-sized models is a more cost-effective strategy than relying on direct use of LLMs for inference in a production setting, which typically faces recurrent API usage costs or requires extensive high-end computational infrastructures for fine-tuning.

## 2. Data

The source data used in this experiment comprise titles and abstracts from a large collection of around 476k research articles in the AECO area published in the time range 2010-2023, retrieved from the OpenAlex[2] open scientific graph database [16] using a set of platform-specific topic filtering tags.

We sampled a test set of around 50 abstracts, pre-processed and sentence split them using Spacy's English transformer pipeline *en_core_web_trf-3.6.1*[3] and finally had them independently annotated by two domain experts using the Brat annotation tool [17], resulting in a total of 314 sentences, 448 entities, 132 relations instances. The inter-annotator positive specific agreement on entity detection ([18][4]) reached a mean F1 score of 0.73, indicating an overall satisfying agreement between the human annotators, although some marginal ambiguity is encountered for such a complex task. We publicly share the current version of the test dataset (called SCIERC-AECO) in the github: https://github.com/zavavan/sperty/blob/main/datasets/scierc_aec/scierc_aec_test.json and plan to release extended versions in the future.

We used two random samples of respectively 3 and 10 sentences as example annotations for the few-shot LLM prompts described below.

## 3. Experimental Setups

The full-stack Relation Extraction task consists of generating, for an input token sequence $X = x_0, ..., x_n$: a) a set of tuples $E = <(x_i, ..x_j), T_e>$ of typed token sub-sequences of $X$, with $0 \leq i, j \leq n$ and $T_e \in T^E$ being a label belonging to the set $T^E$ of entity labels; b) a set $R$ of triples

---

[2]https://docs.openalex.org/
[3]https://github.com/explosion/spacy-models/releases/tag/en_core_web_trf-3.6.1
[4]This corresponds to classical Cohen κ inter-rater agreement, in tasks like NER where the number of negative cases is undefined.

$< h, t, T_r >$ where $h, t \in E$ are, respectively, the entity head and tail of the relation, and $T_r \in T^R$ is a relation label.

As baseline for the RE task we use SpERT (Span-based Entity and Relation Transformer) [19], a span-based model for joint entity and relation extraction. SpERT is a relatively simple approach using the pre-trained BERT for input token representation that classifies any arbitrary candidate token span into entity types, filters non-entities (*None* entity class) and finally classifies all pairs of remaining entities into relations.

By using only sentence-level context representations for sampling positive and negative training examples, the architecture allows single-pass runs through BERT for each sentence, resulting in significant speeding up of training. Despite this sentence-level RE simplification though, SpERT significantly outperforms other joint entity/relation extraction models on SciERC dataset, reaching up to 70.33% micro-average F1 on entity extraction and up to 50.84% micro-average F1 on relation extraction (around 2.5% improvement on both tasks).

We re-trained SpERT on SciERC training set (1861 sentences) using SciBERT (cased) embeddings [20]. When tested over out-of-domain SCIERC-AECO data though, SpERT performance degrades drastically. First row in Table 1 shows Micro-average F1 scores on SCIERC-AECO for entity extraction (NER), relation detection without argument entity classification (RE) and relation detection considering entity classification (RE_w/NEC), respectively.

In order to test few-shot learning capability of LLMs for training data generation, we experiment on schema-constrained instruction prompts sent to the Chat Completion endpoint of the OpenAI *gpt-3.5-turbo-0125* (ChatGPT) API [21]. The context length of the model is approximately 4096 tokens.

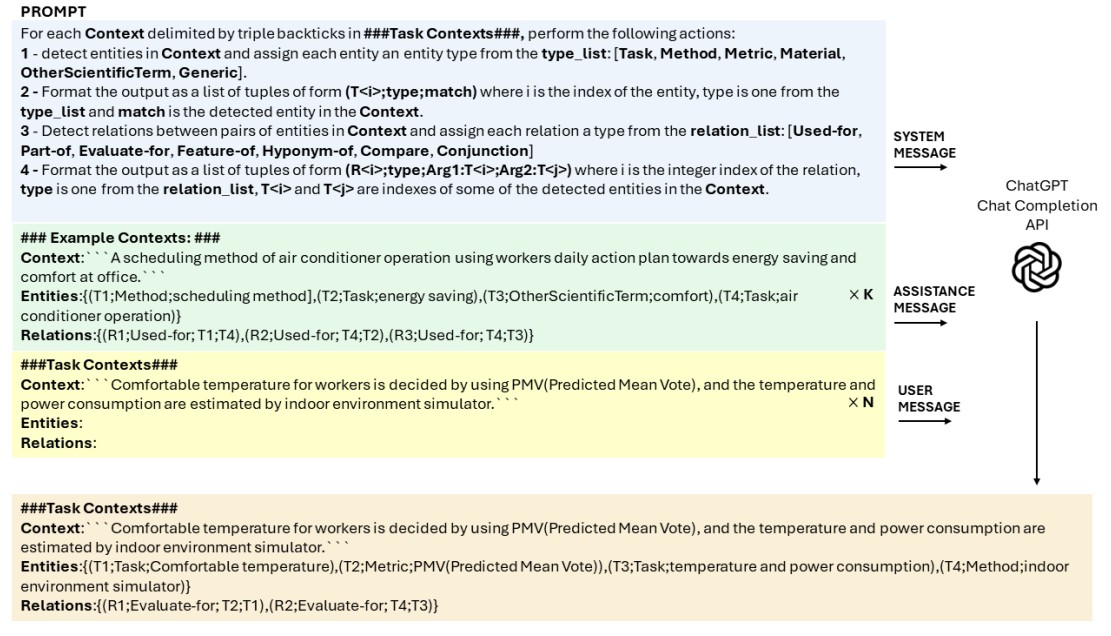

**Figure 2:** Basic structure of the annotation instruction Prompt to ChatGPT, with sample response message (bottom box).

**Table 1**

Micro F1 (%) performance of the baseline SpERT model trained on SciERC dataset and its ChatGPT training data variations generated with different few-shot learning prompt configurations.

| | Method | NER | | RE | | RE_w/NEC | |
|---|---|---|---|---|---|---|---|
| | | $K = 3$ | $K = 10$ | $K = 3$ | $K = 10$ | $K = 3$ | $K = 10$ |
| | SpERT [19] | 17.69 | | 7.05 | | 6.5 | |
| ChatGPT | Schema Prompt | 19.4 | 19.64 | 4.08 | 4.13 | 2.27 | 2.36 |
| | Schema/Descr Prompt | 19.51 | 19.88 | 5.49 | 7.48 | 3.77 | 5.61 |
| | Schema Prompt + SciERC | 19.56 | 22.55 | 7.78 | 11.03 | 5.2 | 7.52 |
| | Schema/Descr Prompt + SciERC | 19.26 | **23.78** | 8.79 | **21.82** | 5.49 | **17.27** |

Figure 2 displays the structure of a schema-based annotation instruction Prompt, consisting of: a) a Task Definition part encoded as System Message and containing a definition of the Task as sequence of annotation actions; b) a set of $K$ sample texts together with their expected annotations, encoded as Assistance message; c) a set of $N$ input text samples, encoded as User Message[5].

We test a few variations of this Prompt schema. Namely, we optionally add plain text descriptions of the Entity and Relation types to increase LLM comprehension of the annotation task. Then, we increment the number of prompt examples to $K = \{3, 10\}$. Full prompt instructions and examples can be found at https://github.com/zavavan/sperty/tree/main/prompts.

All ChatGPT API calls were run with *temperature* parameter set to 0, *top_p* token probability mass sampling set to the default value 1, token *frequency_penalty* and *presence_penalty* set to 0 default value.

For each prompt configuration, we query ChatGPT on 3373 sentences from AECO abstracts, parse ChatGPT response messages into a training dataset and train SpERT with SciBERT (cased) embeddings[6] on it. Finally, we evaluate each model on the SCIERC-AECO test set. We also test for training SpERT with a merge of ChatGPT-generated data with original (out-of-domain) SciERC dataset (3800 sentences).

## 4. Results and Discussion

Table 1 reports Micro-average F1 scores when training with ChatGPT-generated data using the basic instruction Prompt (*Schema Prompt*) and the prompt enriched with schema description (*Schema/Descr Prompt*), for different values of $K$ prompt examples. The last two rows report performance when training on merged ChatGPT-generated and SciERC data.

At a first glance the LLM seems to fully comply with the structural requirements of the annotation task, consistently generating schema-based output. In some cases, it "semantically" manipulates the input text (average 2% occurrence) so as to make the output annotations not directly usable for sequence labeling. For example, from the sentence *"The carbon emissions*

---

[5]Due to OpenAI API maximum request tokens limit, these are sent in batches of 10 sentences each.

[6]We trained for 20 epochs with 0.1 dropout rate. Experiments were run on NVIDIA A100-SXM4-40GB GPU device.

*throughout the entire life cycle of the building have been reduced by 20.99%.*" it generates an entity not anchored in text*(T1;Task;Carbon emissions reduction)*[7].

Overall, the performance level is not outstanding across all configurations, considering that the same model architecture is achieving a F1 measure of 2-3 factors higher when trained on in-domain manually curated data (SciERC) of comparable size. This may be due to the model degrading its generalization performance by learning from noisy labels [22], which is confirmed by observing that the best results are obtained by adding ChatGPT generated labels to curated out-of-domain SciERC labels. By considering only LLM-generated data, most configurations slightly outperform the baseline for NER while only one does it for RE, indicating that this is an harder task for LLM few-shot learning with respect to NER.

Adding explicit Task definitions and increasing the number of few-shot examples both consistently raise the performance with respect to all metrics, with the second finding seemingly in contrast with what reported in [15].

## 5. Conclusions

This contribution presents our currently-ongoing work on the potentials of Large Language Models (LLMs), specifically ChatGPT, for few-shot learning in the context of relation extraction domain adaptation. In particular the study aimed to generate in-domain training data for a Transformer-based relation extraction model within the Architecture, Construction, Engineering, and Operations (AECO) domain by leveraging the in-context learning capabilities of LLMs. The experiments involved using structured prompts and minimal expert annotation to collect training instances from AECO research paper titles and abstracts.

The results indicate that the quality of the LLM-generated annotations may not be sufficient to support domain customization of a RE model from ground up. However, when combined with curated out-of-domain labels it can boost the performance on the new domain significantly.

Overall, the research highlights the potential of using ChaptGPT for optimizing local, lower-sized models, which can be a more cost-effective strategy than relying on direct use of LLMs for inference in production settings.

Future work might include expanding the test set and conducting more extensive tests to further validate the approach, also considering other domains than AECO. In addition, in the future we plan to experiment with GPT-4 for data generation rather that ChatGPT, leveraging its powerful capabilities to improve the quality of synthetic data, such as in the powerful LLaVA multi-modal model [23].

Object of further investigation and experiments will involve also the use of the latest advances in open-source LLMs, such as by employing open-source models like *Mistral-7B-OpenOrca*[8], *Nous Hermes Mixtral*[9], and *Llama-3-70B*[10], to explore their potential in relation extraction tasks. This could involve comparing the performance of these models directly with the current

---

[7]In a few other cases, annotation labels out of the schema are assigned to Entity and Relations.
[8]Mistral-7B-OpenOrca, https://huggingface.co/Open-Orca/Mistral-7B-OpenOrca
[9]Nous Hermes Mixtral, https://huggingface.co/NousResearch/Nous-Hermes-2-Mixtral-8x7B-DPO
[10]Llama-3-70B, https://huggingface.co/meta-llama/Meta-Llama-3-70B

approach, as well as exploring their capabilities in generating high-quality synthetic data for fine-tuning smaller models like SpERT.

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
