# OpenReview forum: "A Few-Shot Approach for Relation Extraction Domain Adaptation using Large Language Models"
_KDD.org/2024/Workshop/DL4KG — DL4KG 2024_

### Official Review · Reviewer_iiyv · 2024-07-01
**A Few-Shot Approach for Relation Extraction Domain Adaptation using Large Language Models**

**Rating:** 7
**Confidence:** 3

**Review:**

The paper presents a novel approach to domain adaptation in relation extraction using large language models (LLMs). The authors propose leveraging the in-context learning capabilities of LLMs to perform schema-constrained data annotation, collecting in-domain training instances for a Transformer-based relation extraction model deployed on titles and abstracts of research papers in the Architecture, Construction, Engineering, and Operations (AECO) domain.

One of the key strengths of this paper is its focus on a specific and relevant problem in knowledge graph generation: the need for domain-specific models that can accurately extract relations from text. The authors provide a detailed description of their approach, including the use of structured prompts and minimal expert annotation. This makes the approach more accessible and applicable to other researchers.

Authors use a large dataset of 476k research articles in the AECO area published between 2010 and 2023, retrieved from OpenAlex. This provides a solid foundation for evaluating the performance of the proposed method. However, it would be helpful to see more details about the preprocessing steps and how the data was prepared for training. Authors do send github link but it would help to see the details.

However, there are some areas where the paper could be improved. For example, the authors state that the method is effective for in-domain manually curated data, but they do not provide enough information about how they evaluated the performance of their method compared to existing approaches. It would be useful to see a more detailed comparison with other methods.

Overall, this paper presents an interesting and promising approach to domain adaptation for relation extraction. While there are some limitations in terms of evaluation and comparison, the idea of the paper for data augmentation and schema-constrained annotation is a valuable contribution to the field. Further work is needed to fully evaluate the effectiveness of the approach and compare it with other methods.

---

### Official Review · Reviewer_BXZo · 2024-07-02
**Interesting use case, would benefit from a deeper analysis and discussion**

**Rating:** 6
**Confidence:** 4

**Review:**

This short paper explores the utilization of in-context learning capabilities of Large Language Models (LLMs) to perform schema-constrained data annotation for generating training instances. These instances are used to train a Transformer-based relation extraction model, specifically applied to titles and abstracts in the Architecture, Construction, Engineering, and Operations (AECO) domain. The study demonstrates the potential for domain adaptation of a science KG generation model using a few-shot learning strategy with minimal expert annotation and evaluates against a baseline using SpERT, a model for joint entity and relation extraction that takes as input pre-trained embeddings.

The paper would benefit from a more detailed explanation of the approach. This includes a clearer step-by-step description of how the schema-constrained data annotation process is conducted, more details on the structured prompts that include Relation and Entity type text definitions, and the exact setup of the few-shot learning strategy. It is also likely that other prompts were tested, but did not perform as well. A brief discussion on this would be interesting.

I understand there is supplementary material covering some of this, but considering the paper is not at the length limit (and that some aspects can be summarized, see below) there should be sufficient space to directly address these issues in the main text.

It would be interesting to understand whether the better performance seen when combining ChatGPT-generated data with original (out-of-domain) SciERC dataset is due to a larger number of training instances (it appears that the size is more than doubled with the merge), or due to some synergistic effect between in-domain and out-of-domain data.

The “Results and Discussion” section is currently underdeveloped. Expanding this section to include detailed reflections on the challenges faced, how they were addressed, and insights gained during the experimentation would provide valuable insight for future research. There is a good potential to present this work as a “lessons learned” use case that may guide others that face similar challenges in other domains.

---

### Official Review · Reviewer_UKvV · 2024-07-04
**The research lacks some solid justification**

**Rating:** 4
**Confidence:** 5

**Review:**

The main problem reading the paper I have is the following: the authors use a LLM to generate training instances that will later be used to train a classifier that performs relation extraction.
I don't understand (or I could not find any justification about it) why the training instances are needed to be generated from a LLM.
If the ultimate goal is to perform relation extraction, why don't the authors use the LLM to directly tackle that task?
Results and discussions of the paper are therefore built on top of this problem. If the authors can find a justification of why they need a training set generated from a LLM with respect to use directly a LLM to solve the relation extraction task then I will be happy to reconsider my score.

---

### Decision · Program_Chairs · 2024-07-09

Accept